# Association between 3-Year Repetitive Isolated Hematuria and eGFR Deterioration in an Apparently Healthy Population: A Retrospective Cohort Study

**DOI:** 10.3390/ijerph191811466

**Published:** 2022-09-12

**Authors:** Mami Ishida, Keiichi Matsuzaki, Hitoshi Suzuki, Yusuke Suzuki, Takashi Kawamura, Yoshinori Marunaka, Taku Iwami

**Affiliations:** 1Department of Preventive Services, School of Public Health, Graduate School of Medicine, Kyoto University, Kyoto 606-8303, Japan; 2Agency for Health, Safety and Environment, Kyoto University, Kyoto 606-8501, Japan; 3Department of Nephrology, Juntendo University Faculty of Medicine, Tokyo 113-8421, Japan; 4Medical Research Institute, Kyoto Industrial Health Association, Kyoto 604-8472, Japan

**Keywords:** persistent hematuria, IgA nephropathy, chronic glomerulonephritis, chronic kidney disease, screening, health checkup

## Abstract

Microscopic hematuria is being increasingly recognized as a major indicator of kidney deterioration. Persistent hematuria may better detect estimated glomerular filtration rate (eGFR) deterioration and potential glomerulonephritis. We conducted a retrospective cohort study to investigate the associations between persistent hematuria: the frequency or consistency of positive dipstick hematuria defined by the preceding 3 years urinalyses, and eGFR deterioration over 5 years and abnormal urinalyses suggesting potential glomerulonephritis (hematuria 1+ or higher, 2+ or higher, proteinuria, and hematuria and proteinuria) 5 years later, among adult participants with positive dipstick hematuria at baseline in a large-scale Japanese health checkup setting (n = 2104). There was no significant association between persistent hematuria and eGFR deterioration over 5 years. The higher the frequency of preceding hematuria, the greater the RR of hematuria 5 years later; RRs of hematuria with preceding thrice, twice, or once hematuria were 3.64 [95% CI, 3.11–4.25], 2.97 [95% CI, 2.52–3.51], or 1.91 [95% CI, 1.58–2.30] for “hematuria 1+ or higher,” and 7.13 [95% CI, 5.17–9.83], 4.26 [95% CI, 3.02–6.02], or 2.23 [95% CI, 1.52–3.27] for “hematuria 2+ or higher”. The presence of both hematuria and proteinuria 5 years later was only associated with preceding thrice hematuria (RR: 2.35 [95% CI, 1.37–4.03]). In conclusion, persistent hematuria for 3 years was associated with hematuria and proteinuria that were suggesting glomerulonephritis, but not associated with eGFR deterioration over 5 years. Multiple dipstick urinalyses over years can add some values to detect potential glomerulonephritis as an early sign of chronic kidney diseases.

## 1. Introduction

Chronic kidney disease (CKD) is a major global health problem because it significantly predisposes patients to end-stage kidney disease (ESKD), cardiovascular diseases (CVDs), hospitalization, and premature death [1,2,3]. CKD was the third major responsible disease for the overall years of life lost, following Alzheimer’s disease and diabetes mellitus [4]. A recent global survey found that 1.2 million people died from CKD worldwide in 2017 [5]. In Japan, the number of CKD patients was reported to be 13.3 million people, equivalent to 13% of adults in 2005 [6]. Despite the widespread prevalence of CKD, it is often undetected in its early stage [7,8], since many patients with early-stage CKD have no recognizable symptoms. Patients therefore remain unaware of the need for treatment in its early stages and may overlook its progression. Therefore, good indicators of CKD risk during its early stages and a screening strategy to identify early-stage CKD would lead to better management through early referral to a nephrologist and reduce the burden of CKD worldwide.

Annual health checkups, including dipstick urinary screening, are mandatory for Japanese people in schools, workplaces, and local communities [9]. Positive hematuria by dipstick is common in these health checkups, accounting for approximately 5–10% of the total population [10]. Hematuria is not specific to CKD, as it is also seen in transient urological diseases such as urinary tract infection. Many people with dipstick hematuria thus do not undergo subsequent detailed examinations to find its cause. However, previous reports from Japan suggested that dipstick hematuria in mass screening is a risk factor for ESKD (odds ratio [OR] 1.38 [95% confidence interval (CI), 1.19–1.61]), mortality (hazard ratio [HR] 1.46 [95% CI, 1.15–1.85]), and cardiovascular diseases (HR 1.83 [95% CI, 1.23–2.71]) in men [11,12,13]. A report from Korea suggested that microscopic hematuria is associated with an increased risk of incidence of CKD in a general population (HR 1.37 [95% CI, 1.04–1.79]) [14]. Microscopic hematuria can be an early sign of CKD due to chronic glomerulonephritis and can result in renal deterioration in decades. Nevertheless, hematuria at a single point is weak in predicting future CKD, whereas hematuria is frequently found for close examinations to be carried out. Therefore, a more sensitive/specific strategy is required to effectively screen for potential CKD

Two previous reports showed that isolated asymptomatic microscopic hematuria had a 41% and 83% chance of becoming persistent, and proteinuria then occurred with a probability of approximately 10%, over a 5–6 year period [15,16]. Because patients with both hematuria and proteinuria can develop progressive glomerulonephritis and consequent CKD, persistent hematuria could be an earlier and more specific risk factor for CKD than hematuria at a single point. Persistent hematuria was a major risk factor for ESKD in an Israeli military health checkup (HR 18.5 [95% CI, 12.4–27.6]) [17]. However, this Israeli cohort consisted of very young adults and only a relatively small proportion (15.4%) had IgA nephropathy (IgAN). These results might therefore differ from those in East Asia; half of all cases of persistent isolated hematuria in Japan and Korea were IgAN [18,19], and 70% of IgAN cases in Japan were discovered by urinary screening in health checkups [9,20]. In other words, persistent hematuria might be more associated with poor renal prognosis in areas with a higher prevalence of IgAN. Identifying dipstick hematuria over years may therefore more efficiently find potential glomerulonephritis, resulting in higher awareness of early-stage CKD.

This study aimed to investigate the association between persistent hematuria over 3 years by dipstick and eGFR deterioration and urinary findings suggesting potential glomerulonephritis in a Japanese health checkup setting.

## 2. Materials and Methods

### 2.1. Study Design and Population

This was a single-center, retrospective, observational study. A total of 410,527 adults undertook the health checkup provided by the Kyoto Industrial Health Association between April 2011 and March 2012 (index health checkup). Among the examinees, we extracted those aged 20–65 years who had hematuria at the index health checkup, whose dipstick urinalyses data in the preceding 3 years were available, and whose serum creatinine data at baseline and 5 years later were available (Figure 1). These ages were selected because they are appropriate for hematuria screening to detect glomerulonephritis. Participants who had proteinuria, kidney dysfunction (estimated glomerular filtration rate [eGFR] < 60 mL/min/1.73 m^2^), nephrological or urological disease history, or menstruation at the index health checkup were excluded.

### 2.2. Persistent Hematuria

Urine blood was tested using a dipstick standardized in Japan [21]. Hematuria was defined as the test result of 1+ or higher in this study. Persistent hematuria was considered as frequent or consistent positive dipstick test for hematuria. We classified the study participants into four groups according to the frequency of hematuria at 3 years before the index health checkup when one’s hematuria was detected: thrice, twice, once, or never (Figure 2a). To investigate the association between persistent hematuria and outcomes more precisely, we also classified the study participants into two groups as per the consistency of hematuria; three or more consecutive hematuria vs. the others (Figure 2b).

### 2.3. Outcomes

The primary outcome was renal deterioration, i.e., reduction in eGFR 5 years later. eGFR was calculated from serum creatinine using the Japanese standard equation as follows [22]:eGFR (Cr) (mL/min/1.73 m^2^) = 194 × Cr^−1.094^ × age^−0.287^ (female: × 0.739)

The secondary outcomes were dipstick urinalyses suggesting glomerulonephritis 5 years later. The urinalyses suggesting glomerulonephritis included the presence of hematuria (1+ or higher: 1+, 2+, or 3+, 2+ or higher: 2+ or 3+), incidence of proteinuria (1+ or higher: 1+, 2+, or 3+), and incidence of both hematuria (1+ or higher: 1+, 2+, or 3+) and proteinuria (1+ or higher: 1+, 2+ or 3+). Additional outcomes were incidence of CKD (eGFR < 60 mL/min/1.73 m^2^), 30% reduction in eGFR, and newly developed glomerulonephritis, ESKD, urological diseases, or hypertension based on the answer to the questionnaire at the 5-year-later health checkup.

### 2.4. Confounding Factors

We collected confounding factors related to hematuria and renal deterioration. Demographic variables including age and sex, clinical variables of body mass index (BMI), systolic/diastolic blood pressure, plasma glucose, serum lipid, and serum uric acid, past and present history of hypertension, diabetes mellitus, hyperlipidemia, hyperuricemia, and urological diseases were obtained using anthropometric measures, laboratory tests, and questionnaires in the index health checkup.

BMI was calculated as weight in kg divided by height in m squared (kg/m^2^) and categorized into four groups: underweight (<18.5), normal weight (≥18.5, <25), overweight (≥25.0, <30), and obese (≥30). Hypertension was defined as systolic blood pressure ≥ 140 mm Hg, diastolic blood pressure ≥ 90 mm Hg, or physician’s diagnosis of hypertension. Diabetes mellitus was defined as blood glucose level ≥ 126 mg/dL or physician’s diagnosis of diabetes mellitus. Dyslipidemia was defined as serum low-density lipoprotein (LDL)-cholesterol ≥ 140 mg/dL, high-density lipoprotein cholesterol < 30 mg/dL, serum triglyceride (TG) ≥ 150 mg/dL, or physician’s diagnosis of dyslipidemia. Hyperuricemia was defined as serum uric acid level ≥ 7.0 mg/dL or physician’s diagnosis of hyperuricemia.

### 2.5. Statistical Analyses

In the descriptive analysis, continuous variables were expressed as mean (standard deviation, SD) and the intergroup difference was tested by Student’s *t*-test for two-group comparison and one-way ANOVA. Categorical variables were expressed as frequency by category and tested by χ2 test or Fisher’s exact test. Trend tests were conducted to assure the unidirectional change in frequency. To investigate the associations between persistent hematuria and urinalyses suggesting glomerulonephritis after 5 years, we calculated the risk ratio of each urinalysis as the risk among participants with persistent hematuria divided by the risk among those without persistent hematuria. We added the subgroup analyses by age or sex on the association between persistent hematuria and the results of four urinalyses after 5 years. Moreover, we performed multiple regression analysis for numerical outcomes and logistic regression analysis for dichotomous outcomes, adjusting for the confounding factors (age, sex, BMI, hypertension, diabetes mellitus, dyslipidemia, hyperuricemia, and baseline eGFR). A complete case analysis was conducted. All tests were two-tailed, and *p* values < 0.05 were considered statistically significant. R version 3.6.1 (The R Foundation for Statistical Computing, Vienna, Austria), and EZR version 1.37 (Saitama Medical Center, Jichi Medical University, Saitama, Japan) [23] were used for all analyses.

### 2.6. Ethical Consideration

All analyses were conducted after the complete de-identification of the data in accordance with the Declaration of Helsinki and the national ethics guidelines of Japan, then individual informed consent was waived. This study protocol was approved by the institutional review boards of Kyoto University (Ethics Code No: R1602) and Kyoto Industrial Health Association (18-0003). There are no conflicts of interest to be declared.

## 3. Results

### 3.1. Study Participant

During the study entry period, a total of 410,527 persons underwent the health checkups provided by the Kyoto Industrial Health Association. Of these, 22,541 aged 20 to 65 years and exhibiting hematuria were extracted. Among them, 2104 participants whose preceding 3-year urinalyses data and serum creatinine data (at baseline and five years later) were available were enrolled in the present study (Figure 3).

### 3.2. Main Analysis

Table 1 shows the baseline characteristics of the participants stratified into four groups by their frequency of hematuria (thrice, twice, once, never) until the index health checkup. Frequent hematuria was associated with increased age, female sex, greater BMI, severe hematuria, and reduced eGFR at the baseline.

Figure 4 shows the association between the frequency of hematuria and renal deterioration. The eGFR reduction was 2.28 (8.66) mL/min/1.73 m^2^, 2.67 (9.31) mL/min/1.73 m^2^, 2.16 (8.14) mL/min/1.73 m^2^ or 2.37 (8.83) mL/min/1.73 m^2^ in the group of the frequency of hematuria: thrice, twice, once or never, respectively. There was no significant association between the frequency of hematuria and the 5-year changes in eGFR (*p* value 0.4).

Figure 5 shows the prevalence of four urinalyses suggesting glomerulonephritis after 5 years. The prevalence of hematuria 1+ or higher was 83.5%, 68.2%, 43.8%, and 23.0% in each frequency of hematuria: thrice, twice, once, and never, respectively (Figure 5a). The prevalence of hematuria 2+ or higher was 47.7%, 28.5%, 14.9%, and 6.69% (Figure 5b). The prevalence of proteinuria was 7.81%, 5.37%, 2.75%, and 5.60% (Figure 5c), and the prevalence of hematuria and proteinuria was 7.22%, 3.50%, 1.83%, and 3.07% (Figure 5d) in each frequency of hematuria: thrice, twice, once, or never, respectively. The greater the frequency of preceding hematuria, the greater the probability of glomerulonephritis-suggesting urinalyses after 5 years, except for incidence of proteinuria and incidence of both hematuria and proteinuria in the group with one preceding hematuria finding.

Table 2 shows the association between the frequency of hematuria and four urinalyses, expressed by risk ratio. The risk ratio of the presence of hematuria 1+ or higher were 3.64 [95% CI, 3.11–4.25], 2.97 [95% CI, 2.52–3.51], and 1.91 [95% CI, 1.58–2.30] in the group of the frequency of hematuria: thrice, twice, or once, compared with the group of never hematuria, respectively. The odds ratio of the presence of hematuria 1+ or higher, analyzed by multiple logistic regression adjusted for age, sex, BMI, hypertension, diabetes mellitus, dyslipidemia, hyperuricemia, and baseline eGFR were 16.68 [95% CI, 12.53–22.40], 7.20 [95% CI, 5.41–9.63], and 2.54 [95% CI, 1.93–3.36] in the group of each frequency of hematuria: thrice, twice, or once, compared with the group of never hematuria, respectively (Table 3, Figure 6). Similarly, the risk ratios of the presence of hematuria 2+ or higher were 7.13 [95% CI, 5.17–9.83], 4.26 [95% CI, 3.02–6.02], and 2.23 [95% CI, 1.52–3.27], and the odds ratios of the presence of hematuria 2+ or higher were 12.79 [95% CI, 8.92–18.84], 5.52 [95% CI, 3.75–8.30], and 2.39 [95% CI, 1.56–3.69] in the group of each frequency of hematuria: thrice, twice, and once, respectively. The incidence of proteinuria was only associated with the group with three preceding hematuria findings, as shown by the odds ratio of 1.66 [95% CI, 1.04–2.70], although its risk ratio was not significantly high, which is 1.39 [95% CI, 0.91–2.14]. Similarly, the incidence of both hematuria and proteinuria was only associated with the group with three preceding hematuria findings, as shown by the risk ratio of 2.35 [95% CI, 1.37–4.03] and the odds ratio of 2.89 [95% CI, 1.65–5.30]. Regarding the subgroup analyses by age, a similar result was provided to the main results regarding the association between the frequency of hematuria and presence of hematuria after 5 years (Appendix A). Regarding the subgroup analysis by sex, almost the same results were provided as the main results about the association between the frequency of hematuria and all four urinalyses after 5 years except for the risk ratios of the presence of hematuria 2+ or higher; the RR in women were higher than that in men (Appendix A). There were no significant associations between the frequency of hematuria and outcomes, including incidence of CKD, 30% eGFR reduction, newly developed hypertension, glomerulonephritis, and urological diseases after 5 years. No incidence of ESKD was noted.

### 3.3. Additional Analysis

We added the analysis of another classification of persistent hematuria according to the consistency of hematuria (Figure 2b, Appendix A); three or more consecutive hematuria vs. the others. There was no significant association between the consistency of hematuria and the 5-year changes in eGFR (Appendix A). The consecutive hematuria was significantly associated with the four urinalyses suggesting glomerulonephritis (Appendix A, Appendix A). The results were almost similar with the main results.

## 4. Discussion

This study indicated that persistent hematuria over 3 years was associated with urinary abnormalities suggesting glomerulonephritis, including the presence of hematuria, incidence of proteinuria, and presence of hematuria and proteinuria 5 years later, though not associated with eGFR deterioration. These associations mean that persistent hematuria detected by multiple dipstick urinalyses over years could add some values to the screening potential of glomerulonephritis as an early sign of CKD. There was almost no difference between ages and sexes regarding the presence of hematuria 5 years later. To our knowledge, this is the first study to demonstrate the value of multiple annual dipsticks to find potential glomerulonephritis in a health checkup setting. This approach would strengthen the cooperation between primary care providers and nephrologists.

Our study has two main strengths. First, our cohort participants were recruited at a large health checkup center in Japan where as many as 400,000 examinees including employees, independent business persons, housewives, and retired persons dwelling in Kyoto Prefecture and its vicinity undertook health checkups in a year. The participants are therefore likely to be highly representative of the general population in Japan. Second, our findings can be applied to any other Japanese setting because the health checkup systems and screening tests are thoroughly standardized in Japan [24] and the semi-quantitative dipstick urinalysis was quite familiar to healthcare providers.

We could not indicate that persistent hematuria was associated with eGFR deterioration over 5 years. There are some plausible reasons for this. First, the follow-up period of 5 years might be too short to assess one’s renal deterioration based on eGFR. Glomerulonephritis exhibiting hematuria without proteinuria would be at its very early stage and be far from excretive dysfunction. The period from the onset to advanced CKD of IgAN, most common glomerulonephritis, is usually decades [25,26,27]. In this case, at least 10 years would be necessary to assess a reduction in eGFR. Second, non-renal diseases such as bladder stone or inflammation that usually do not develop CKD were not completely eliminated in this study, even though one’s past and present disease history was inquired in the questionnaire of the health checkups.

Most primary chronic glomerulonephritis, especially IgAN, is known to first manifest itself as asymptomatic proteinuria and/or hematuria [28,29]. IgAN exhibits a variety of clinical manifestations due to its heterogeneous etiology. The two most common clinical presentations are asymptomatic microscopic hematuria and progressive inflammatory kidney disease [30]. Asymptomatic hematuria with minimal proteinuria was previously regarded as having a favorable prognosis, but in East Asia, 30%–40% of these cases developed hypertension and significant proteinuria within 7–9 years [31,32]. Moreover, persistent microscopic hematuria defined as time-averaged hematuria by urinary sediment at baseline and during follow-up was identified as a risk factor for the progression of IgAN [33]. Our results, showing that persistent dipstick hematuria was associated with “hematuria and proteinuria” 5 years later, suggest that isolated hematuria, if it continues, was likely to develop proteinuria. Furthermore, persistent hematuria can imply the continuance of active inflammation due to very early-stage chronic glomerulonephritis, which may develop into ESKD in the following decades. Potential CKD could therefore be identified more efficiently by the screening of persistent hematuria rather than hematuria at a single point. To achieve this, it is more realistic to use multiple annual dipstick urinalysis than urinary sediment, as the examination of urinary sediment is not suitable for large populations in a health checkup setting.

How to determine “persistent hematuria” by dipstick urinalysis has been under debate. Persistent hematuria was previously defined as five or more red blood cells per high-power field in a urine specimen obtained on the three separate occasions on different days [17]. Based on this study, we assured that the harm of hematuria would depend on the frequency or continuity of dipstick hematuria and defined both frequent and consistent positive dipstick hematuria as persistent hematuria in this study. We analyzed the two patterns based on the different definition of persistent hematuria, as shown Figure 2a,b. These two patterns of persistent hematuria provided the similar results of the association between persistent hematuria and urinary findings suggesting potential glomerulonephritis. We regarded the frequency pattern as the main analysis because the risk of persistent hematuria can be interpreted more finely. These analyses straightened our hypothesis that persistent hematuria could be screening potential glomerulonephritis as an early sign of CKD.

This study has several limitations. First, an accurate diagnosis of glomerulonephritis by renal biopsy and urological diseases by close examinations was not necessarily carried out for the participants. Second, the outcome measurement was eGFR based on serum creatinine obtained at only two points, which may not accurately assess renal deterioration. eGFR can be less reliable in a population with normal kidney function [22], and 5 years may be too short to investigate renal survival, as mentioned above. Third, in this study, we defined persistent hematuria by dipstick over 3 years as an exposure. However, participants with persistent hematuria over 3 years could include those with diseases generally known to have a good renal prognosis, such as thin glomerular basement membrane syndrome. However, this would have reduced the risk ratios in our results, and this bias would not be a serious problem. To target a population with future high-risk CKD, it is important to know the onset and duration of hematuria. An analysis of the participants’ lifelong results would more specifically identify participants with elevated CKD risk. Fourth, we selected the participants with preceding three year-urinalyses and serum creatinine to define persistent hematuria and evaluate kidney function, and we could not analyze all participants with hematuria. However, the reasons why we could not collect preceding three years and serum creatinine were not because of their health condition, but mostly because of their environmental reasons not related to their health problems, such as transfer of residences or companies; therefore we considered that there would not be a large difference between the participants of this study and people with hematuria who were excluded in this study. Lastly, the participants were limited to those who visited the same heath checkup center for 8 years, which could lead to some selection biases. However, for most participants, annual health checkup was mandated, and this selection bias would be small.

## 5. Conclusions

Persistent hematuria over 3 years was associated with urinary abnormalities suggesting glomerulonephritis. Multiple dipstick urinalyses over years can add some values to detect potential glomerulonephritis as an early sign of CKD.

## Figures and Tables

**Figure 1 ijerph-19-11466-f001:**
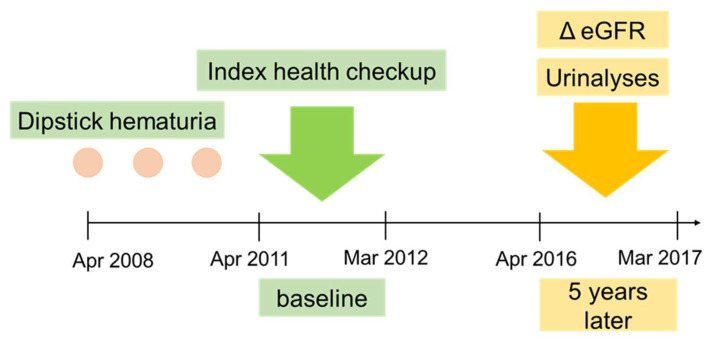
Patient timeline showing timing of enrolment and data collection. Participants with dipstick hematuria at baseline (April 2011 through March 2013) with available three annual dipstick results for the previous 3 years and serum creatinine results at baseline and at the 5-year mark were included in the study. Outcomes were assessed according to estimated glomerular filtration rate (eGFR) and urinalyses.

**Figure 2 ijerph-19-11466-f002:**
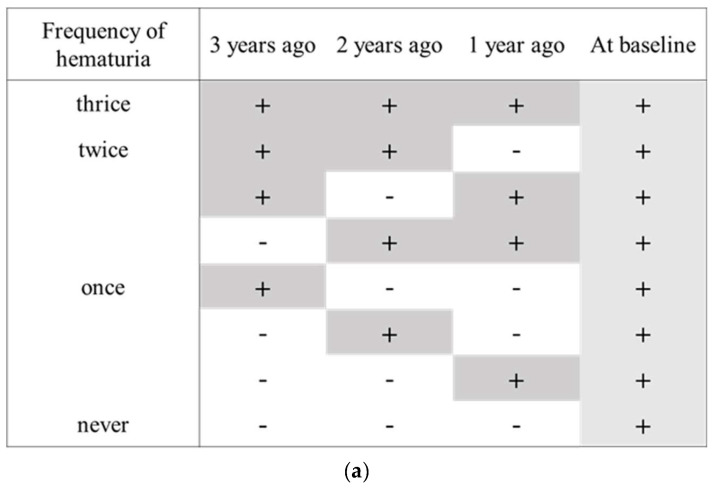
(**a**): Patterns of persistent hematuria defined by the frequency of hematuria. (**b**): Patterns of persistent hematuria defined by the consistency of hematuria. Three or more consecutive hematuria was defined as persistent hematuria and is shown by gray area.

**Figure 3 ijerph-19-11466-f003:**
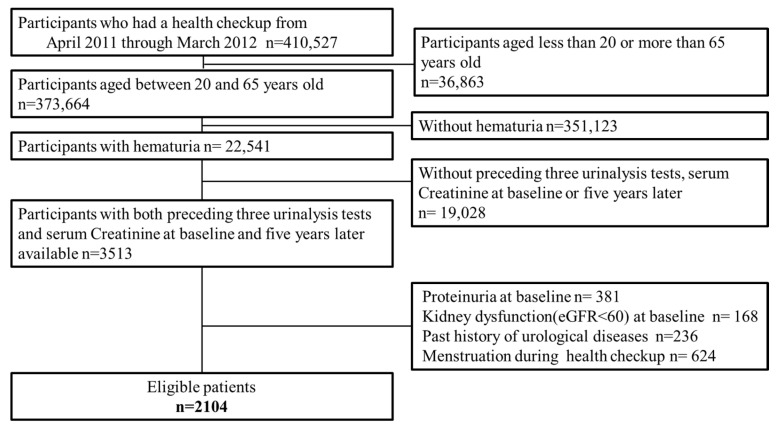
Participants flow.

**Figure 4 ijerph-19-11466-f004:**
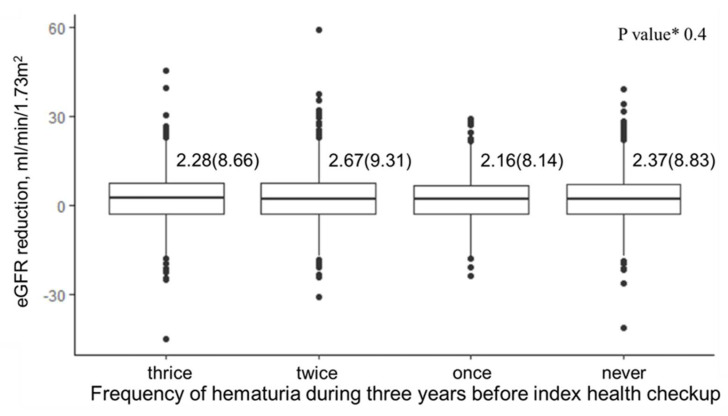
eGFR reduction for 5 years according to the frequency of hematuria. * Adjusted for age, sex, BMI, hypertension, diabetes mellitus, dyslipidemia, hyperuricemia, baseline GFR.

**Figure 5 ijerph-19-11466-f005:**
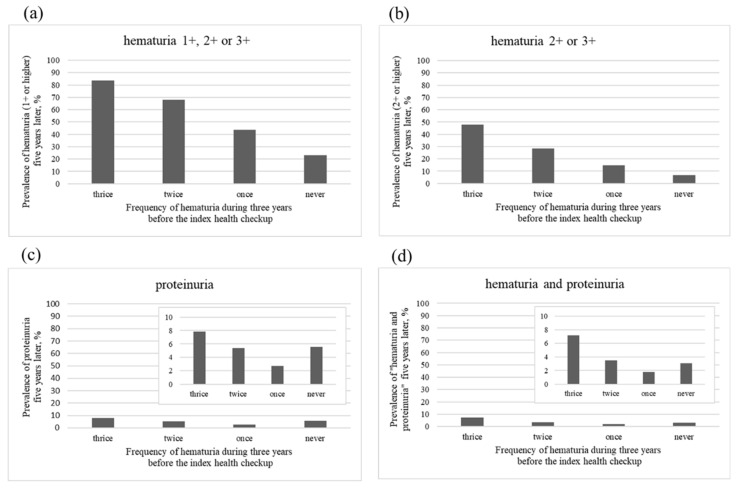
Prevalence of four urinary findings suggesting glomerulonephritis in four groups that is stratified by the frequency of hematuria during three years before the index health checkup. (**a**): hematuria 1+ or higher: 1+, 2+ or 3+; (**b**): hematuria 2+ or higher: 2+ or 3+; (**c**): proteinuria; (**d**): hematuria and proteinuria.

**Figure 6 ijerph-19-11466-f006:**
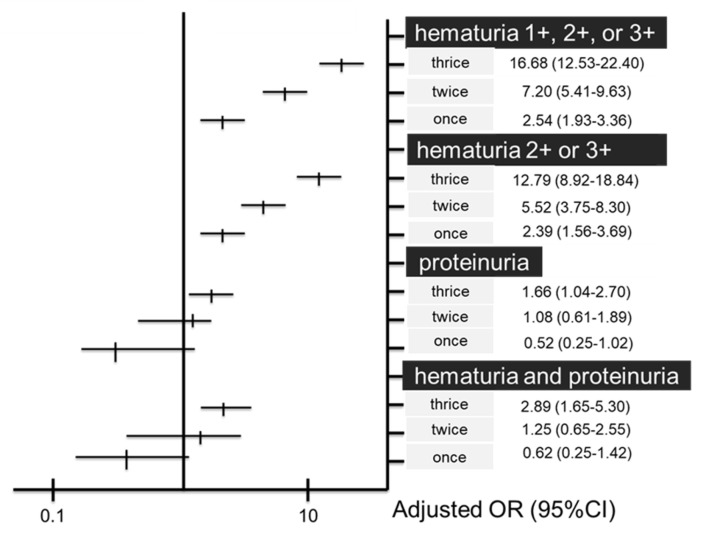
Associations between the frequency of hematuria and four urinary findings suggesting glomerulonephritis 5 years later, expressed as odds ratios; multiple logistic regression analysis. Reference is the group whose frequency of hematuria is never.

**Table 1 ijerph-19-11466-t001:** Baseline characteristics according to the frequency of hematuria.

	Total (n = 2104)	Thrice (n = 680)	Twice (n = 429)	Once (n = 440)	Never (n = 555)	*p* Value
Age, years, mean (SD)	48.3 (8.22)	50.3 (7.79)	48.0 (8.20)	47.6 (8.12)	46.7 (8.36)	<0.001
Male, n (%)	1138 (54.1)	335 (49.3)	222 (51.7)	233 (53.0)	348 (62.7)	<0.001
BMI, n (%)						0.03
<18.5	238 (11.3)	82 (12.1)	50 (11.7)	61 (13.9)	45 (8.1)	
≥18.5, <25	1511 (71.8)	498 (73.3)	308 (71.8)	303 (68.9)	402 (72.4)	
≥25, <30	306 (14.6)	90 (13.3)	61 (14.2)	68 (15.5)	87 (15.7)	
≥30	48 (2.3)	9 (1.3)	10 (2.3)	8 (1.8)	21 (3.8)	
Dipstick hematuria, n (%)						<0.001
1+	1348 (64.1)	315 (46.3)	303 (70.6)	334 (75.9)	396 (71.4)	
2+	639 (30.4)	313 (46.0)	118 (27.5)	88 (20.0)	120 (21.6)	
3+	117 (5.6)	52 (7.7)	8 (1.9)	18 (4.1)	39 (7.0)	
Serum creatinine, mg/dL, mean (SD)	0.740 (0.130)	0.732 (0.129)	0.735 (0.132)	0.729 (0.129)	0.752 (0.140)	0.02
eGFR, mL/min/1.73 m^2^, mean (SD)	80.0 (11.9)	78.38 (11.7)	79.99 (12.3)	80.92 (11.0)	81.36 (12.5)	<0.001
Hypertension, n (%)	651 (31.0)	232 (34.2)	127 (29.7)	118 (26.8)	174 (31.4)	0.06
Diabetes mellitus, n (%)	302 (14.4)	83 (12.2)	64 (14.9)	68 (15.5)	87 (15.7)	0.3
Dyslipidemia, n (%)	869 (41.4)	296 (43.5)	167 (39.0)	185 (42.1)	221 (39.9)	0.4
Hyperuricemia, n (%)	289 (13.8)	95 (14)	49 (11.4)	54 (12.3)	91 (16.4)	0.1

Abbreviations: BMI, body mass index; eGFR, estimated glomerular filtration rate; Missing values: BMI: 1 (thrice), hypertension: 4 (thrice 2; twice 1; never 1), diabetes mellitus: 2 (once), dyslipidemia: 3 (twice 1; once 1; never 1), hyperuricemia: 3 (twice 1; once 1; never 1).

**Table 2 ijerph-19-11466-t002:** Associations between outcomes and the frequency of hematuria, expressed by risk ratios.

	Risk Ratio		
	Thrice	Twice	Once
Dipstick hematuria			
1+, 2+ or 3+	3.64 (3.11–4.25)	2.97 (2.52–3.51)	1.91 (1.58–2.30)
2+ or 3+	7.13 (5.17–9.83)	4.26 (3.02–6.02)	2.23 (1.52–3.27)
Dipstick proteinuria	1.39 (0.91–2.14)	0.96 (0.57–1.62)	0.49 (0.26–0.94)
Hematuria and proteinuria	2.35 (1.37–4.03)	1.14 (0.58–2.26)	0.60 (0.26–1.37)
CKD, eGFR < 60	1.22 (0.79–1.89)	0.61 (0.33–1.11)	0.87 (0.51–1.47)
GFR 30% reduction	1.09 (0.25–4.84)	2.16 (0.52–8.97)	1.26 (0.26–6.22)

Abbreviations: CKD, chronic kidney disease; eGFR; estimated glomerular filtration rate. Reference is the group whose frequency of hematuria is never.

**Table 3 ijerph-19-11466-t003:** Associations between outcomes and the frequency of hematuria, expressed by adjusted odds ratios.

	Multivariate Odds Ratio *		
	Thrice	Twice	Once
Dipstick hematuria			
1+, 2+ or 3+	16.68 (12.53–22.40)	7.2 (5.41–9.63)	2.54 (1.93–3.36)
2+ or 3+	12.79 (8.92–18.84)	5.52 (3.75–8.30)	2.39 (1.56–3.69)
Dipstick proteinuria	1.66 (1.04–2.70)	1.08 (0.61–1.89)	0.52 (0.25–1.02)
Hematuria and proteinuria	2.89 (1.65–5.30)	1.25 (0.61–2.55)	0.62 (0.25–1.42)
CKD, eGFR < 60	0.79 (0.47–1.35)	0.83 (0.45–1.51)	0.94 (0.50–1.72)
GFR 30% reduction	1.31 (0.27–7.11)	2.44 (0.56–1.26)	1.33 (0.24–7.54)

Abbreviations: CKD, chronic kidney disease; eGFR; estimated glomerular filtration rate. Reference is the group whose frequency of hematuria is never. * Logistic regression analysis, adjusted for age, sex, BMI, hypertension, diabetes mellitus, dyslipidemia, hyperuricemia, baseline eGFR.

## Data Availability

The data underlying this article cannot be shared publicly for the privacy of individuals that participated in the study. The data will be shared on reasonable request to the corresponding author.

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
