# Peer review of "Association between 3-Year Repetitive Isolated Hematuria and eGFR Deterioration in an Apparently Healthy Population: A Retrospective Cohort Study"

_ijerph, 2022, doi:10.3390/ijerph191811466_

Round 1

Reviewer 1 Report

The authors have conducted a retrospective cohort study (defined as  "a large-scale (n=2,104 adults) Japanese health checkup setting" ) to investigate the associations between persistent (once, twice or thrice) hematuria (defined by the preceding 3 years urinalyses) and eGFR deterioration over 5 years and abnormal urinalyses suggesting potential glomerulonephritis (hematuria 1+ or higher, 2+ or higher, proteinuria, and hematuria and proteinuria) 5 years later .

There was no significant association between persistent hematuria and eGFR deterioration over 5 years. The higher the frequency of preceding hematuria, the greater the RR of hematuria 5 years later. Presence of both hematuria and proteinuria in 5 years later was only associated with preceding thrice hematuria (RR: 2.35 [95% CI, 1.37–4.03]).

The authors conclude that persistent hematuria for 3 years was associated with hematuria and proteinuria, a finding more suggestive of glomerulonephritis, but not associated with eGFR deterioration over 5 years. They conclude that multiple dipstick urinalyses over years can add some values to detect potential glomerulonephritis as an early sign of chronic kidney disease. 

This is an important additional contribution to previous reports  by Vivante et al (NEJM, JAMA), who showed that even isolated asymptomatic persistent microhematuria at age < 18 yrs (w/o associated proteinuria) was a significant risk factor for end stage kidney disease, decades  later. 

The authors admit in their introduction, that even though screening with dipstick urinalysis is mandatory in Japan, and is a risk factor for the detection of CKD or ESRD, the rate of false positivity from such test (for example by urinary tract infection or menstruation) makes its use as an efficient screening method - problematic. Persistent hematuria may be a better screening method, especially in areas of the world with higher incidence of IgA nephropathy, like east Asia.  

The authors concentrate on 410,527 adults who undertook the health checkup provided by the Kyoto Industrial Health Association between April 2011 and March 2012 (index health checkup). Among them they extracted those aged 20–65 years, who had hematuria at the index health checkup, whose dipstick urinalyses data in the preceding 3 years were available, and whose serum creatinine data at baseline and 5 years later were available. Participants with dipstick hematuria at baseline (April 2011 through March 2013), previous available three annual dipstick results  and serum creatinine results at baseline and at the 5 year were included in the study. Outcomes were assessed  according to estimated glomerular filtration rate (eGFR) and urinalysis. 

Authors provide precise definitions of hematuria (persistent, once, twice, thrice, consistent and non consistent), possible glomerulonephritis and chronic kidney disease.

Although hematuria was found in 22541 out of 373664 eligible patients (age 20-65), only 3513 had 3 previous urine analyses, as requested in the inclusion criteria and only 2104 were eligible patients . 

Specific critiques:

1)  Table 1: the study population's mean age was close to 48 years, but a ~ 3 years difference is seen between the once Vs thrice hematuria groups. Is this consistent with the guidelines for hematuria screening in Japan?  When is screening for hamaturia recommended ? This should appear in the Methods section.  

2) The authors have excluded pts who tested positive for blood but did not have urine dipstick data  from previous 3 yrs . How does  this group (which should be more representative of the general population in th eKyoto area) differ from the selected group in terms of their baseline demographic and clinical data as well as their outcome?  

3) In the Methods section, the authors do not specify how was the data accessed and whether a Human Experimentation Committee approval was obtained. In addition, the completeness of data is not clear. 

Minor critique:  

1) Line 267-70: there is a whole paragraph at the beginning of the Discussion section which includes instructions for the authors and should be omitted. 

Reviewer 2 Report

Thank you for presenting how persistent hematuria monitoring over, with utilizing the dipstick could be used to detect potential early signs of CKD or other glomerular diseases. 

Overall the paper and data look sound. I only have a couple comments.

Intro: Please define hematuria for the purposes of this study. 

Discussion: The first paragraph can be removed as I think it was mistakenly left during the writing process. It describes what should be included in the discussion from the template.
